# Change Detection and Impact of Climate Changes to Iraqi Southern Marshes Using Landsat 2 MSS, Landsat 8 OLI and Sentinel 2 MSI Data and GIS Applications

**Bassim Mohammed Hashim [1]** , **Maitham Abdullah Sultan [1]** , **Mazin Najem Attyia [2]** , **Ali A. Al Maliki [1]** and **Nadhir Al-Ansari [3],\***

[1] Ministry of Science and Technology, AL Mustansiriyah University, Baghdad 10001, Iraq; bassim_saa22@yahoo.com (B.M.H.); maitham_nlt@yahoo.com (M.A.S.); alyay004@mymail.unisa.edu.au (A.A.A.M.)

[2] Iraqi Atomic Energy Commission, Baghdad PO Box 765, Iraq; mazinmd724@yahoo.com

[3] Department of Civil, Environmental and Natural Resources Engineering, Lulea University of Technology, 97132 Lulea, Sweden

\* Correspondence: nadhir.alansari@ltu.se; Tel.: +46-0046920491858

**Abstract:** Marshes represent a unique ecosystem covering a large area of southern Iraq. In a major environmental disaster, the marshes of Iraq were drained, especially during the 1990s. Since then, droughts and the decrease in water imports from the Tigris and Euphrates rivers from Turkey and Iran have prevented them from regaining their former extent. The aim of this research is to extract the values of the normalized difference vegetation index (NDVI) for the period 1977–2017 from Landsat 2 MSS (multispectral scanner), Landsat 8 OLI (operational land imager) and Sentinel 2 MSI (multi-spectral imaging mission) satellite images and use supervised classification to quantify land and water cover change. The results from the two satellites (Landsat 2 and Landsat 8) are compared with Sentinel 2 to determine the best tool for detecting changes in land and water cover. We also assess the potential impacts of climate change through the study of the annual average maximum temperature and precipitation in different areas in the marshes for the period 1981–2016. The NDVI analysis and image classification showed the degradation of vegetation and water bodies in the marshes, as vast areas of natural vegetation and agricultural lands disappeared and were replaced with barren areas. The marshes were influenced by climatic change, including rising temperature and the diminishing amount of precipitation during 1981–2016.

**Keywords:** Iraqi marshes; climate change; satellites data

## 1. Introduction

The marshes in the southern part of Iraq are among the most prominent phenomena in the region that formed within the sedimentary plain forming a natural balance between the Tigris, Euphrates rivers and Shatt al-Arab leading to the Arabian Gulf. The marshes represent an integrated ecosystem that dates back more than 5000 years and occupied a large area of southern Iraq [1]. The marshes and the surrounding areas in southern Iraq were characterized by the availability of water and the climatic viability of growing economic crops and raising buffalo. However, it has been severely damaged by the drought. Fish are one of the most important elements of livestock in the marshes and are characterized by their abundance. The marshes are also a source of income for many fishermen living in the area. The areas of the marshes are the most important environments for breeding; it is also the habitat of

many birds, many of which have migrated from different regions of the world, such as Siberia and northern Europe, especially in winter and spring. The marshes contain reeds and papyrus, which are used for the manufacture of various stocks, some of which are animal feed [2].

The draining of the marshes in the second half of the 20th Century, and particularly in the 1990s, severely disrupted the hydrological regime of the marshes. Large dams in the upper reaches of the Euphrates and Tigris started to change the water distribution throughout the basin since the mid-20th Century, strongly impacting downstream water use [3]. Turkey first launched its Southeast Anatolia development project, including 22l dams and 19 hydropower plants, in 1977, and rebalanced it in 1989. Iran started large-scale water management projects on the Karun and Karkheh rivers (the important tributaries of Hawiza marsh), in the mid-1990s [4]. Since 2003, local inhabitants began to reflood some of the marshes, often in an uncontrolled and haphazard fashion [5]. Because of the breaching of levees and dams and coincidental plentiful rain in the following two years, the marshes superficially recovered and regained about 55% of their former extent [6]. This recovery was transient, however.

Following a drought in 2008–2009, marsh extent declined again and only slightly recovered in the winter of 2009/2010 [7]. The drying up of the marshes has had a significant impact on climate change, in the southern region of Iraq at the very least if the impact is not wider. This means that the area the marshes and its adjacent areas have clear differences in the elements of the climate, resulting from the destruction of the environment and natural resources in the region [8]. Climate change caused by the increase in greenhouse gases in the atmosphere has significantly influenced the water balance by causing changes in evapotranspiration rates, temperature, and precipitation. These changes have had a negative impact on water resource availability [9]. As an important part of the Earth's water cycle, land surface water bodies, such as rivers, lakes, and reservoirs, are irreplaceable for the global ecosystem and climate system. Surveying land surface water bodies and delineating their spatial distribution has a great significance in understanding hydrology processes and managing water resources [10]. At present, remote sensing has become a routine approach for many fields, including geography, land surveying, and most Earth Science disciplines (for example, hydrology, ecology, meteorology, oceanography, glaciology, geology); it is also used for land surface and water body monitoring, because the acquired data can provide macroscopic, real-time, dynamic and cost-effective information, which is substantially different from conventional in situ measurements [11].

Landsat has recently been used to undertake long term monitoring of surface water bodies at global and regional scales [12]. Landsat-8 acquires global moderate-resolution measurements of the Earth's surface in the visible (VIS), near-infrared (NIR), short wave infrared (SWIR) and thermal infrared (IR) [13]. The Sentinel 2A satellite was successfully launched on 23 June 2015, as part of the European Copernicus program. Sentinel 2 carries an innovative wide-swath, high-resolution (up to 10 m for four spectral bands), multispectral imager (MSI) with 13 spectral bands; this is going to offer unprecedented perspectives on our land and vegetation [14]. Many studies have dealt with the use of Landsat in the study and monitoring of marshes and wetlands. Abdul Jabbar et al. [15] used Landsat images in three different intervals: MSS in 1973, TM in 1990 and ETM in 2000. These were used in the change detection method and in calculating the normalized difference vegetation index (NDVI) to recognize the vegetation cover and determined as a percentage from the total coverage area of the marshes. The results have shown there are large changes that took place between 1973 and 2000 in land cover and land use. The barren land increased; while the water bodies decreased drastically. The work of Amani et al. [16] deals with the spectral characteristics of five wetland types, including bog, fen, marsh, swamp, and shallow water, in a pilot site in Newfoundland were analyzed. This study used the data acquired by the two satellites, Sentinel-2A and Landsat-8. According to the analyses, the best spectral bands for wetlands' discrimination and classification were selected. Then, the optimum bands were inserted into an object-based Random Forest algorithm to classify wetlands in the study area. The overall classification accuracy was 84% with a Kappa Coefficient of 0.77.



The objectives of this research are: (1) to extract NDVI values and using Landsat 2, Landsat 8 and Sentinel 2 satellites data of the marshes; (2) to use supervised classification to identify land and water surface classes to detect changes in the marshes during 1977–2017; (3) to determine the potential impacts of climate changes through the study of the annual average of the maximum temperature and the precipitation in different areas in the marshes for 1981–2016; (4) to compare between the results of Landsat 2 and Landsat 8 with Sentinel 2 to find the best tool to detect changes to the cover of land and water.

## 2. Materials and Methods

### 2.1. Study Area

The marshes spread the swamps in the south of the sedimentary plain of Iraq, are as a triangle; the urban areas of Amara, Nassirya, and Basra are situated on their heads, (Figure 1). The most important of these marshes are Hawiza Marsh, which intersects with the Iranian border, the central marshes between two provinces Misan and Thi Qar, and Hammar Marsh, which lies south of the central marshes on the Euphrates river. The marshes are located in a basin area, where the lands descend from different directions. The land in the marshland is quite flat and low at the same time, but the height of the area does not exceed 2.5 m, above the sea level, and the rise in sea level is in Hammar Marsh. The marshes are located in the subtropical zone, which is characterized by a hot, humid climate in summer and cold in the winter with precipitation. The maximum temperature in summer ranges between 35–50 °C and in January it reaches 0 °C [17]. In the winter, the area is subject to low pressure, causing winds from the northeast. In the summer, northwestern wind blow over the marshes. During the last decade of the 20th century, there was a clear increase in temperature and low relative humidity. Drying the marshes negatively affected the climate of the region. As a result of high temperatures, atmospheric pressure decreases and wind speed increases in the region. The lower relative humidity in the air was due to the receding of the marshes, which is the main source of air supply with water vapor. The rising dust phenomenon in the marshes has also increased as a result of the drought of a large area of land, which was immersed in water [18].

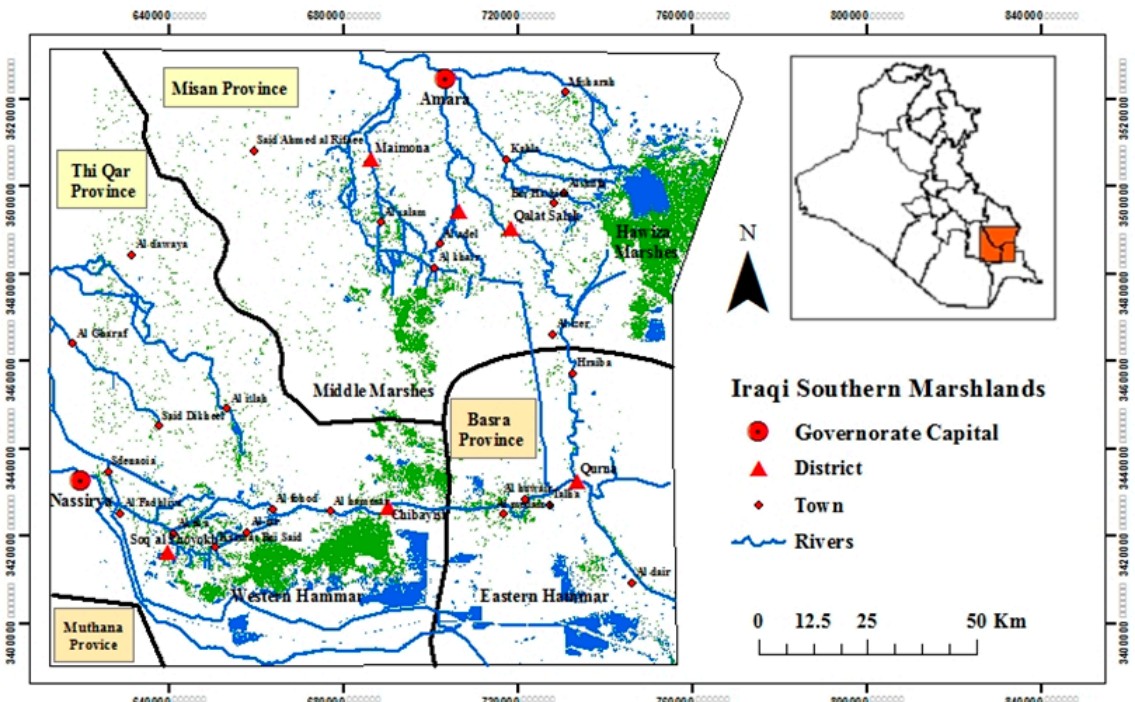

**Figure 1.** Marshlands in southern Iraq.

*2.2. Available Data*

The digital elevation model (DEM) derived from the Shuttle Topography Radar Mission (STRM3) [19], used to determine the elevation of the Earth's surface relative to sea level of the marshes in southern Iraq and create a hydrological analysis using Arc GIS 10.4.1. In this study, imagery from three satellites, Landsat 2, 8 and Sentinel 2, was used. Table 1 [20,21] shows the characteristics of each satellite. Figure 2 shows images used in the current study with the date of each.

**Table 1.** Characteristics of Landsat 2, 8 and Sentinel 2 satellites [20,21].

| Satellite | | Sepectral Bands (μm) and Spatial Resolution (m) | Swath Area (Km) and Temporal Resolution | Launch and work period | Sensor |
|---|---|---|---|---|---|
| Landsat-2 | Band 4 | 80 m Blue (0.5 to 0.6 μm) | 170 km x 185 km 16 days | 1975–1983 | MSS (Multispectral Scanner System) |
| | Band 5 | 80 m Green (0.6 to 0.7 μm) | | | |
| | Band 6 | 80 m Red (0.7 to 0.8 μm) | | | |
| | Band 7 | 80 m NIR (0.8 to 1.1 μm) | | | |
| | Band 1 | 30 m Visible (0.43 to 0.45 μm) | 170 km x 185 km 16 days | | OLI (Operation Land Imager) |
| | Band 2 | 30 m Visible (0.53 to 0.59 μm) | | | |
| | Band 3 | 30 m Red (0.64 to 0.67 μm) | | | |
| | Band 4 | 30 m Blue (0.5 to 0.6 μm) | | | |
| | Band 5 | 30 m Blue (0.5 to 0.6 μm) | | | |
| | Band 6 | 30 m SWIR (1.57–1.65 μm) | | | |
| | Band 7 | 30 m SWIR (2.11–2.29 μm) | | | |
| | Band 8 | 30 m Panchromatic (0.5–0.68 μm) | | | |
| | Band 9 | 30 m Cirrus (1.36–1.38 μm) | | | |
| | Band 10 | 100 m Thermal (10.6–11.19 μm) | | | |
| Lansat-8 | Band 11 | 100 m Thermal (11.5–12.51 μm) | | 2013–2015 | MS1 (Multispectral Scanner System) |
| | Band 1 | 60 m Visible (0.433 to 0.453 μm) | | | |
| | Band 2 | 10 m Blue (0.458 to 0.523 μm) | 290 km x 290 km 10 days | | |
| | Band 3 | 10 m Green (0.543 to 0.573 μm) | | | |
| | Band 4 | 10 m Red (0.650 to 0.680 μm) | | | |
| | Band 5 | 20 m Red Edge (0.698 to 0.713 μm) | | | |
| | Band 6 | 20 m Red Edge (0.733 to 0.748 μm) | | | |
| | Band 7 | 20 m Red Edge (0.765 to 0.785 μm) | | | |
| | Band 8 | 10 m NIR (0.930 to 0.950 μm) | | | |
| | Band 8A | 20 m Red Edge (0.855 to 0.875 μm) | | | |
| Sentinel-2 | Band 9 | 60 m NIR (0.930 to 0.950 μm) | | | |
| | Band 10 | 60 m Cirrus (1.365–1.385 μm) | | | |
| | Band 11 | 20 m SWIR (1.565–1.655 μm) | | | |
| | Band 12 | 20 m SWIR (2100–2.280 μm) | | | |

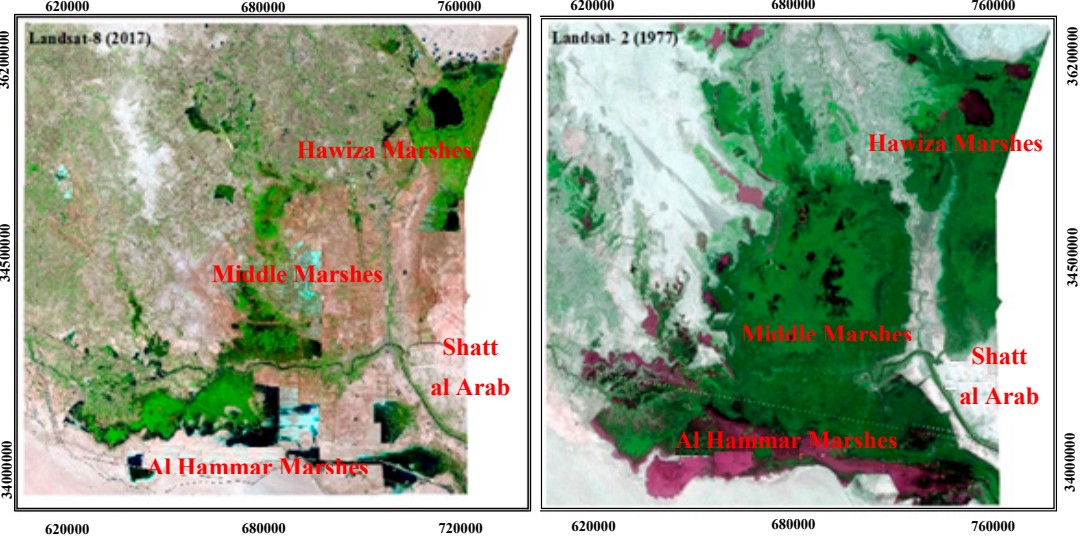

**Figure 2.** *Cont.*

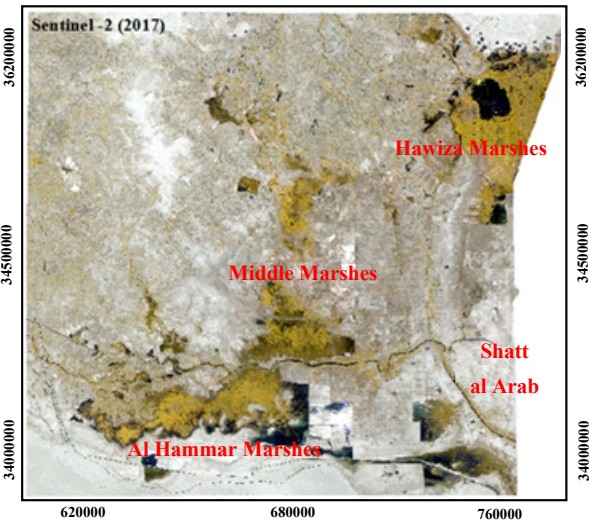

**Figure 2.** Images used in the current study.

*2.3. Methodology*

The study methodology included the following steps:

1.  Acquisition of raw data for satellites Landsat 2, 8 and Sentinel 2 of the Marshlands area of the US Geological Survey site [22]. Landsat 2 images were obtained on 10 June 1977, and 3 August 1977; because there were no serial images. Landsat 8 images were also 2 October 2017 and 11 October 2017. Sentinel 2 was obtained on 2 and 3 October 2017.
2.  Merging the spectral bands of images for the study area to obtained composite images.
3.  Mosaic for the images of each satellite separately to get one large image while keeping all the data in it.
4.  Cutting the study area (Clip) from the resulting image of the mosaic.
5.  Calculation of the NDVI for each satellite image separately. This indicator is one of the most useful methods for monitoring and distinguishing vegetation from water. It depends on the relationship between the spectral bands NIR and Red band R, due to the high reflectivity of the plant in the NIR range and its low reflectivity in R (Red band) for the VIS spectrum. The value of the NDVI is between −1 and +1, where the value from −1 to 0 represents water and other surface types, while the plant represents values from 0 to +1. NDVI is calculated according to the following Equation [23]:

$$NDVI = \frac{NIR(band) - R(band)}{NIR(band) + R(band)} \tag{1}$$

6.  The ArcGIS 10.4.1 program is used to perform the supervised classification process, which is the division of the image according to their digital elements (DN) values. This requires the availability of samples representing the categories of classification in certain areas of the image and the production of land cover maps for the study area. The calculation of the number of changes in the area of water and plants in the marshes of southern Iraq for 1977 and 2017

**3. Results and Discussions**

*3.1. DEM*

ArcGIS 10.4.1 was used to determine the DEM of the marshes in southern Iraq, Figure 3. The slope of the Earth's surface is directly from the northern, northwestern and southwestern areas of the marshes (represented by the direction of arrows in the figure above) towards the central and Hammar marshes, which are the lowest area in the study area.

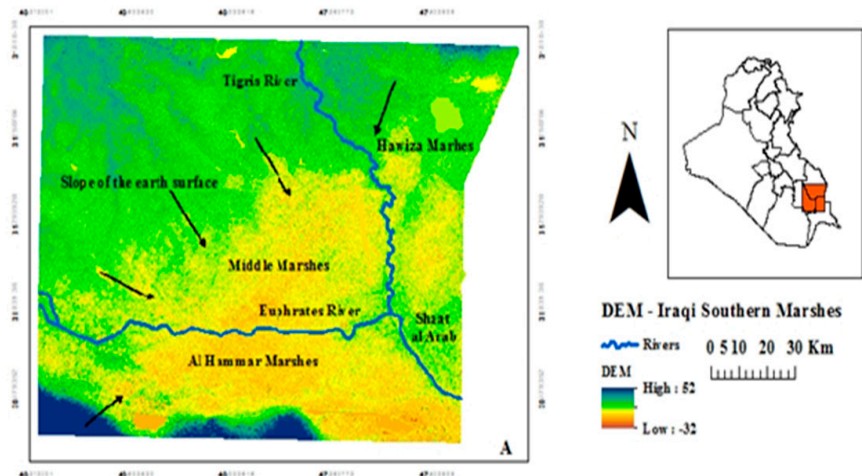

**Figure 3.** Digital elevation model (DEM) for marshes of southern Iraq.

## 3.2. NDVI

ArcGIS 10.4.1 was used to calculate the NDVI values for the three satellite images of the southern Iraqi Marshes for 1977 and 2017, Figure 4. The NDVI values of Landsat 2 images in 1977 were large. The areas of green color in the landscape were wide and extended on a large area within the marshes and on their borders. They represent the natural and agricultural vegetation, and their value ranges from 0 to +1. While the blue color represents the water bodies of the marshes at that time, and their values are less than 0. The 10 m resolution of Sentinel 2 was more accurate in showing vegetation and water for NDVI values than in Landsat 8 in 2017, although the two images were captured at about the same time. The reason was the range of NDVI in Sentinel 2 image ranges between (−1 to 0.84). This allows for slight changes in the difference between R and IR bands, and showing vegetation areas, shallow and deep-water levels in it, compared to Landsat-8 image, in which the value of NDVI ranged from (−0.27 to 0.57). When comparing the NDVI values between 1977 and 2017, observe the deterioration of the vegetation cover and water bodies in the marshes. Large areas of natural vegetation and agricultural land have disappeared, and have been replaced by dry arid lands.

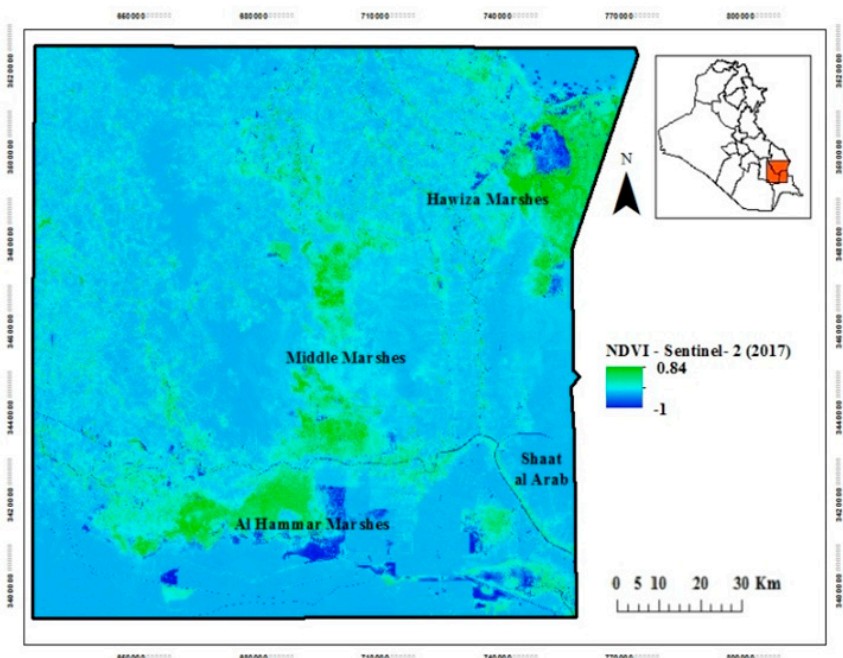

**Figure 4.** *Cont.*

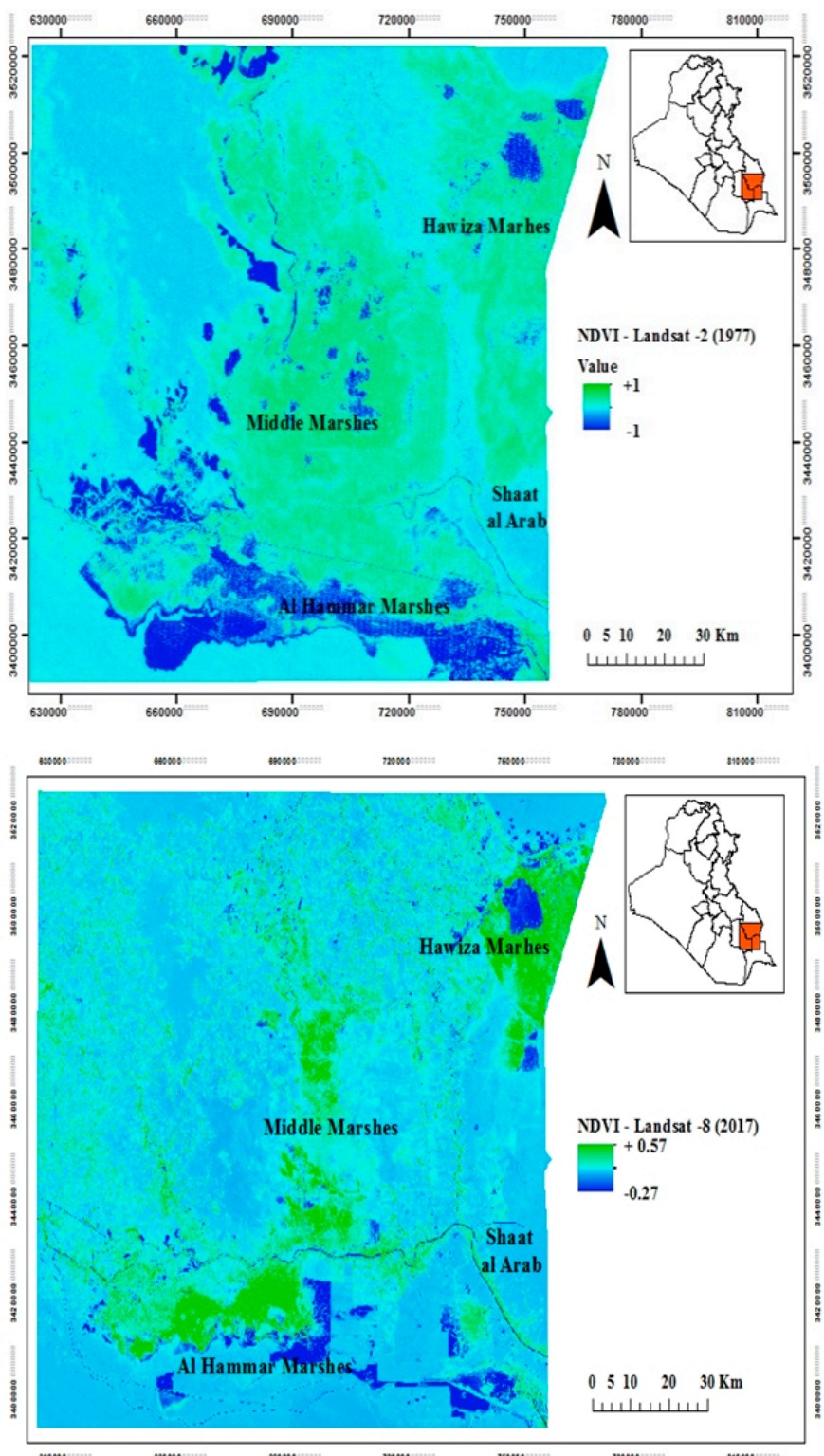

**Figure 4.** Normalized difference vegetation index (NDVI) values for images of (a) Landsat 2 in 1977, (b) Landsat 8 in 2017 and (c) Sentinel 2 in 2017 for southern Iraqi Marshes sowing the differences between the years 1977 and 2017.

### 3.3. Supervised Classification

Displaying the results of the supervised classification for images of three satellites to southern marshes for 1977 and 2017, Figure 5 showed the large variation in the area of the water bodies and the vegetation within 40 years. The area of water bodies in the image of Landsat 2 in 1977, Figure 5A was 3173.92 km$^2$, especially the large expansion for Hammar Marsh. While the area of natural plants reached to 5898.54 km$^2$, it represented the biggest area within the Hawiza and central marshes. The agricultural areas were centered around Hawiza, and the central marshes were 4600.95 km$^2$, on branches of Tigris River in Misan province and branches of Euphrates River within Thi Qar province. The barren area reached to 4257.31 km$^2$ in 1977. Note from Figure 5B illustrates the supervised classification of Landsat 8, that the area of water bodies in marshes has shrunk in 2017 to 753.79 km$^2$, converted to a barren area. As well as, area of natural plants shrinks to 1486.16 km$^2$, agriculture areas to 3806.72 km$^2$. In other hands, the area of barren lands increased to 11,883.47 km$^2$ compared to 1977. For comparison with the image of Landsat 8, the results of supervised classification to Sentinel 2 in 2017, Figure 5C, showed that water area was 544.62 km$^2$, and the agriculture area reached to 3275.38 km$^2$, less than their values in Landsat-8. Meanwhile the area of the natural plant was 1539.86 km$^2$ and barren areas were 12,596.24 km$^2$. The difference in areas of land cover and land use between the images of Landsat-8 and Sentinel 2 came back to clear contrast in the recorded spectral reflectivity for land surface classes (water, natural plant, agriculture and barren lands) in the marshes region, especially within the spectral bands Red Edge 5, 6, 7, 8A, NIR 8 and 9 in Sentinel 2. These bands are useful for moisture land mapping and provide information to vegetation and water monitoring for marshes. There are also the spectral bands SWIR 11 and 12, in Sentinel 2, that corresponding to SWIR 6 and 7 bands in Landsat 8, Table 1. The bands SWIR 11 and SWIR 6 to distinguish water bodies, compared to SWIR 12 and SWIR 7, due to its sensitivity for plants and soil moisture content. While the thermal bands 10 and 11 are in Landsat-8 only, they are used to distinguish shallow water areas. Therefore, the area of water bodies in the image of Landsat 8 bigger than in Sentinel 2, although the two images were taken in the same year. Figure 6 shows a comparison between areas of land cover classes in Iraqi southern marshes for three satellite images in 1977 and 2017. The decreases in water, agricultural lands and natural plants conversely lead to increases in barren areas during 1977–2017. The drying up of marshes since the 1980s and conversion to barren area, destroying the unique ecosystem of marshes, represented bigger environmental disaster, changed features of life and nature in it, which led to a large migration of its residents and neighboring villages. High temperature and evaporation rates accompanied low water revenues of the marshes, as well as decreased precipitation. All these reasons have affected the environment of the marshes and their areas and changed their environment significantly. Despite an inundation of the marshes by water, it has not reached half of its previous area in the 1970s.

The drying processes are not solely responsible for the deterioration of the ecosystem of the marshes. There are many factors which have contributed to this deterioration over the years, including the poor water management of the marshes after the re-inundation, and, affecting Iraq and the marshes in particular, global climate change. This is evident in the change in the annual maximum temperature averages and precipitation, Figure 7, which shows the annual maximum temperature average in the period of 1981–2016, for some cities in the area of the marshes in southern Iraq (Chibayish, Shatra, Amara, Souq al Shoyokh, Maimona, Nasseriya, Qurna and Qalat Saleh). Figure 8 illustrates the annual precipitation average for the same time period and sites. The above two figures show that there has been a clear change in the maximum temperature that has risen in all sites, indicating that the marshes are affected by climate change. The trend line in marshes for the last 40 years shows a weak decrease in precipitation only in 2 cases of 8 stations; but in all cases, recent years were very dry, which has had a significant impact on Iraq's available water imports and reduced water supply to the marshes.

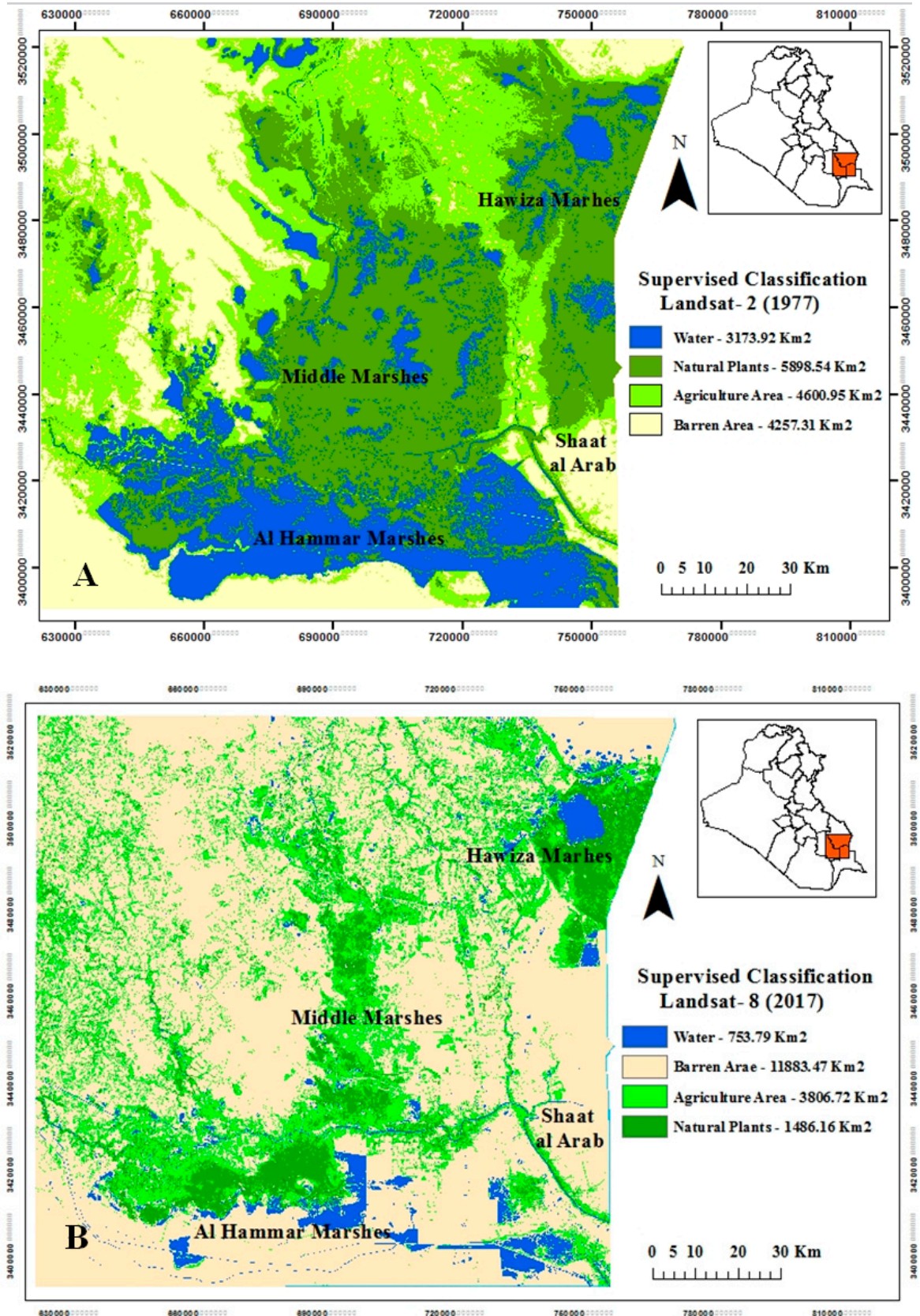

**Figure 5.** *Cont.*

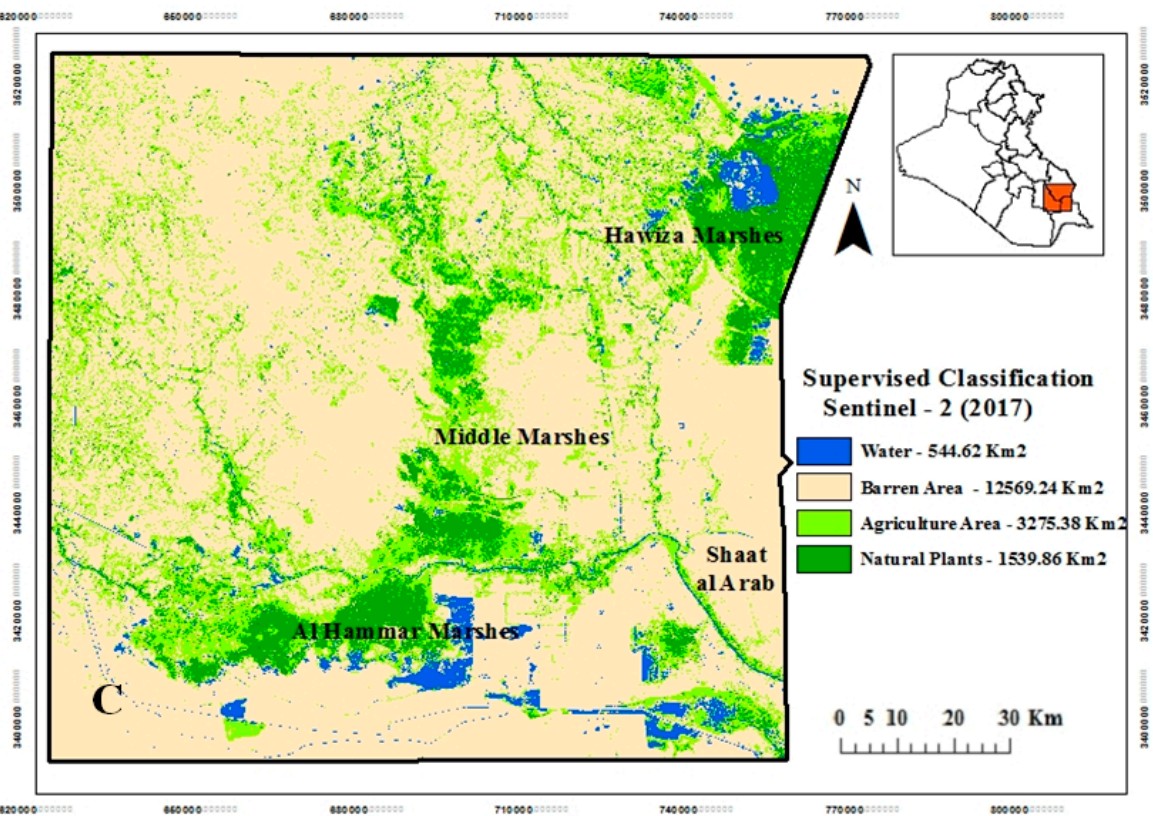

**Figure 5.** Supervised classification for Iraqi southern marshes: (**A**) Landsat 2 in 1977, (**B**) Landsat 8 in 2017 and (**C**) Sentinel 2 2017.

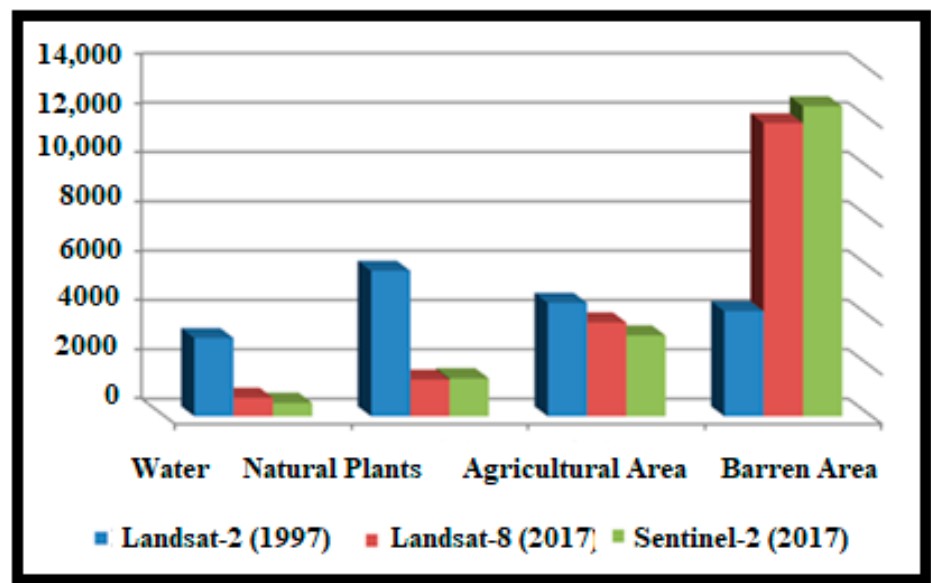

**Figure 6.** Comparison between areas of land covers classes in Iraqi southern marshes for three satellite images in 1977 and 2017.

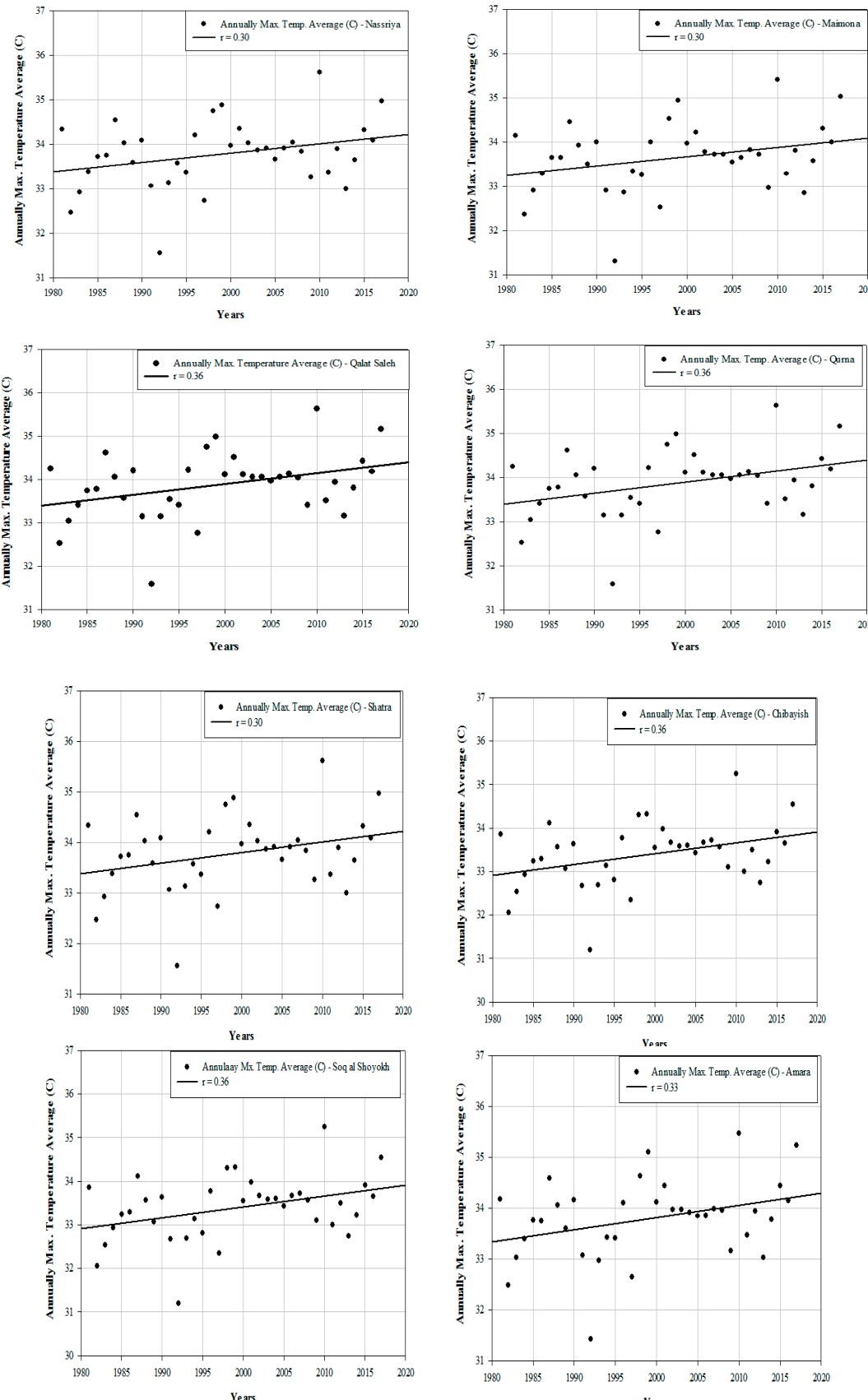

**Figure 7.** Changes of the Annual maximum temperature average in period of 1981–2016.

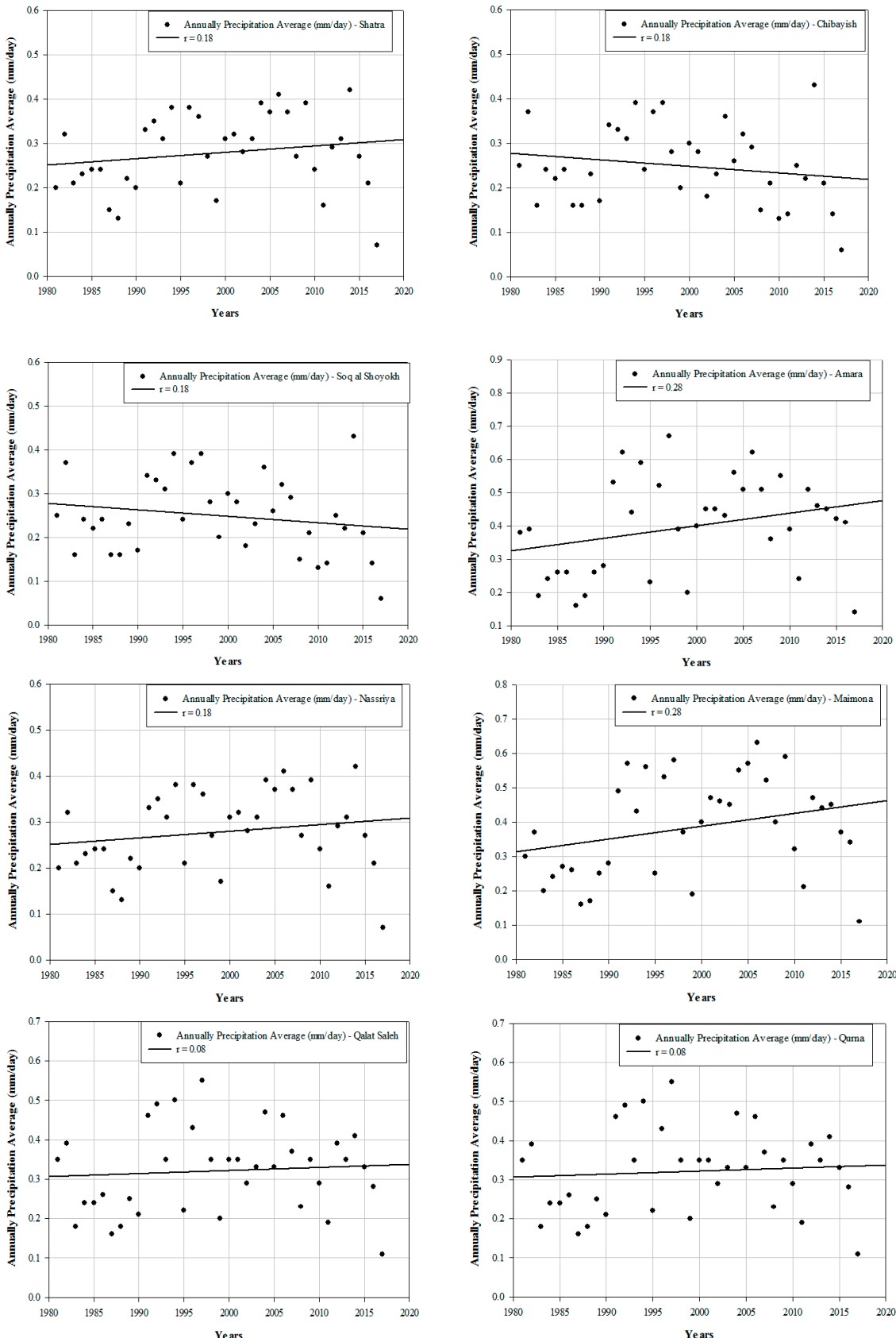

**Figure 8.** Changes of the Annual precipitation average in period of 1981–2016.

## 4. Conclusions

The results of the NDVI calculation of the Iraqi southern marshes for the years 1977 and 2017, showed the deterioration of vegetation cover and water bodies in it. Large areas of natural vegetation and agricultural land disappeared and were replaced by dry barren lands. The drying processes are not solely responsible for the deterioration of the ecosystem of the marshes. Several factors have contributed to this deterioration over the years, including the poor water management of the marshes. Also, the marshes have been affected by climate change, as a result of the increase of an annual average of maximum temperature, accompanied by a decrease in the amount of precipitation. However, satellite data can be used in the periodic monitoring of marshes and water bodies, because it has high potential to detect changes that have occurred. It is important to develop plans to improve the development of the marshes, and to restore them to what they were, because the marshes are very important environmentally and economically to Iraq.

**Author Contributions:** B.M.H., M.A.S., M.N.A. and A.A.A.M. did the methodology, software, field and lab investigations, validation and writing the first draft. N.A.-A. did the discussions, writing the first and final draft.

**Funding:** No external funds were received.

**Conflicts of Interest:** The authors declare no conflict of interest.

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
