# Peer review of "Change Detection and Impact of Climate Changes to Iraqi Southern Marshes Using Landsat 2 MSS, Landsat 8 OLI and Sentinel 2 MSI Data and GIS Applications"

_applsci, doi:10.3390/app9102016_

Round 1

Reviewer 1 Report

It is an interestic regional study.

Author Response

Responding to Peer Reviewer Comments: 

Dear Editor

We would like to thank you and the reviewers for their comments which improved the paper.

Best regards.

Reviewer 1:

Comments and Suggestions for Authors

English level is very poor, author used very long sentences, which is not a good approach in scientific paper. Data and methodology is not properly describe (sources of data, projection, removing errors in data, color combination of NDVI, field data collection and in last accuracy assessment). There is no proper discussion.

Line 65: not only for land surface water; from wikipedia: Remote sensing is the acquisition of information about an object or phenomenon without making physical contact with the object and thus in contrast to on-site observation, especially the Earth. Remote sensing is used in numerous fields, including geography, land surveying and most Earth Science disciplines (for example, hydrology, ecology, meteorology, oceanography, glaciology, geology); it also has military, intelligence, commercial, economic, planning, and humanitarian applications.

Line 75 remove Sentinel 2

Line 109: precisely, which years

Line 114:  why erosion, it is not described before?

Line 164 the reviewer suggested use hillshading

line181: that explains itself using 10 m resolution

line 243: you should combine all 8 station in one box

line 248: the trendline for the last 40 years shows only in 2 cases of 8 stations a weak decrease in precipitation, but in all cases the last years are very dry

Responding to Reviewer 1

Most of the revisions prompted by reviewer2 comments are minor and require no further explanation than what appears throughout the text, other corrections are in order below:

The sentences was recognized ( as yellow color between lines 64-67)

(Sentinel 2)was removed from text: line 77

Sentence corrected as yellow color in line 114

Word of erosion was replaced to drought(line 119)

No change

Sentences was corrected as reviewer suggestion, as in yellow color line 185

Figure 7 was modified according to reviewer comment.

The sentence was change according to reviewer comment. As lines 253-255 with yellow color

Reviewer 2 Report

English level should be improved, author used very long sentences, which is not a good approach in scientific paper. Data and methodology is not properly describe (sources of data, projection, removing errors in data, color combination of NDVI, field data collection and in last accuracy assessment). There is no proper discussion. 

Author Response

Responding to Peer Reviewer Comments: 

Dear Editor

We would like to thank you and the reviewers for their comments which improved the paper.

Best regards.

We would like to thank you. We have improved the paper as you suggested.

Thank you again.

Best regards.

Authors
